# Irreversible and Self-Healing Electrically Conductive Hydrogels Made of Bio-Based Polymers

**DOI:** 10.3390/ijms23020842

**Published:** 2022-01-13

**Authors:** Ahmed Ali Nada, Anita Eckstein Andicsová, Jaroslav Mosnáček

**Affiliations:** 1Centre for Advanced Materials Application, Slovak Academy of Sciences, Dubravska Cesta 9, 845 11 Bratislava, Slovakia; aanada@ncsu.edu; 2Pretreatment and Finishing of Cellulose Based Textiles Department, National Research Centre, Giza 12622, Egypt; 3Polymer Institute, Slovak Academy of Sciences, Dubravska Cesta 9, 845 41 Bratislava, Slovakia; anita.andicsova@savba.sk

**Keywords:** electrically conductive hydrogel, conjugated polymers, self-healing hydrogel, renewable polymers

## Abstract

Electrically conductive materials that are fabricated based on natural polymers have seen significant interest in numerous applications, especially when advanced properties such as self-healing are introduced. In this article review, the hydrogels that are based on natural polymers containing electrically conductive medium were covered, while both irreversible and reversible cross-links are presented. Among the conductive media, a special focus was put on conductive polymers, such as polyaniline, polypyrrole, polyacetylene, and polythiophenes, which can be potentially synthesized from renewable resources. Preparation methods of the conductive irreversible hydrogels that are based on these conductive polymers were reported observing their electrical conductivity values by Siemens per centimeter (S/cm). Additionally, the self-healing systems that were already applied or applicable in electrically conductive hydrogels that are based on natural polymers were presented and classified based on non-covalent or covalent cross-links. The real-time healing, mechanical stability, and electrically conductive values were highlighted.

## 1. Introduction

Natural polymers are often made of polysaccharides and proteins [1] with a huge variety in chemical structures, physical, and biological properties. Polysaccharides such as cellulose [2,3], chitosan [4,5,6], alginate [7], gum Arabic [8], starch [9,10], carrageenan [11,12], and hyaluronic acid [13] have been developed to different applications [14,15,16] due to their biodegradability, biocompatibility, less inflammatory response, producing eco-friendly by-products, and low cost in some cases, such as cellulose [17]. According to the hydrophilic nature of polysaccharides, they show great potentials to form hydrogels.

Hydrogels can be defined as a state of matter between solid and liquid states that are made of cross-linked systems that contain fluids and show no flow in the steady-state condition. These networks can be obtained via either physical or chemical cross-linking bonds in the presence of a medium that fills the gaps and pores of these matrices. Accordingly, hydrogels can absorb water up to as much as 1000 times of its dry weight owing to the hydrophilic functional groups in the polymer, but do not dissolve owing to the cross-links between the polymer chains [18]. Synthetic polymers such as poly(vinyl alcohol) [19], poly(ethylene oxide) [20], poly(ethylene glycol), and poly(acrylamide) [21] can form hydrogels as well with higher mechanical strength than that of natural polymers. However, natural polymers are still preferred due to their biocompatibility and biodegradability.

The production of electrically conductive hydrogels emerged a few decades ago to obtain hydrogels that conduct electricity and to open up new applications for hydrogels. Biopolymers such chitosan [22], cellulose [23], chitin [24], alginate [25], and hyaluronic acid (HA) [26,27] inherently act as insulators. Therefore, such conductive hydrogels have been fabricated mainly by incorporating electrically conductive mediums to the hydrogel networks during or after the gel formation.

In general, there are two types of hydrogels, i.e., irreversible and self-healing (reversible) hydrogels that are based on the nature of the cross-links that occurred between the polymer chains. Each type of hydrogel opens new avenues for applications that span from water treatments to tissue engineering. 

A self-healing hydrogel is a soft material that can absorb and retain water in three-dimensional networks and can automatically recover its mechanical strength after suffering an external mechanical damage [28]. Based on this characteristic, self-healing hydrogels, which are basically made of polymeric matrices, have been used in numerous applications to increase reliability, prolong the life time, and reduce material wastes [29].

With the tremendous technological development in semi/superconductors that relies on electrically conductive mediums such as conductive polymers [30,31,32,33], carbon-based fillers [34,35], or metallic particles [36,37], the need for imparting self-healing properties into the conductive substrates has become a necessity. In self-healing and electrically conductive hydrogels, self-repairing behavior is not limited to restoring the mechanical property but also to restoring the electrical conductivity. Hydrogels that are conducting electricity showed an outstanding performance in bio-applications such as electro-stimulated drug delivery systems [38,39,40], biocompatible neural tissue engineering materials [41,42], and wearable strain sensors [43]. Moreover, it has been reported that electro-conductive self-healing hydrogels are able to completely restore the original mechanical strength after the healing process unlike other self-healing mechanisms [44]. Typically, the self-healing efficiency of non-conductive hydrogels attains up to 90% [45], 80% [46], or less of the original mechanical strength.

The potential of tuning both the mechanical strength and the electrical conductivity values of the self-healing hydrogels offers a wide diversity in applications that spans from ultrasensitive and soft hydrogels for strain sensors [47] to very tough hydrogels for articular cartilage replacement [48]. More recently, a conductive self-healing hydrogel has been used as a binder in a silicon-based anode in lithium-ion batteries. Such a new matrix showed a great potential to maintain the mechanical integrity of the silicon anodes during the frequent charge-discharge cycle compared to regular binding agents [49].

Here we have reviewed both the irreversible as well as the self-healing electrically conductive hydrogels, which at least partially are made of natural polymers. In the first part, electric conductive media, especially conductive polymers, which can be potentially synthesized from renewable resources were reviewed. In the second part, the review on irreversible hydrogels was especially focused on hydrogels that contained conductive polymers as a conductive medium, as, unlike the other conductive media, the conducive polymers can be synthesized in already formed hydrogel. The preparation methods of conductive hydrogels that are based on these conductive polymers were reported by observing their electrical conductivity values by Siemens per centimeter (S/cm). Subsequently, the review of all the so far reported self-healing electrically conductive hydrogels that are based on natural polymers will be provided. The self-healing mechanisms of hydrogel substrates were classified based on covalent or non-covalent bonds. Non-covalent bonding mechanisms (hydrogen-bonding, hydrophobic interactions, guest-host interactions, metal-ligand, and polymer-ions) and covalent bonding mechanisms (Schiff’s base reaction, boron-ester bonds, disulfide bonds, and Diels-Alder reaction) were surveyed for the hydrogels. The real-time healing, healing conditions, mechanical stability, and electrical conductivity values were highlighted.

## 2. Electrically Conductive Media

Electrically conductive substrates have been produced by incorporating conductive fillers such as metallic particles, carbon-based additives (carbon black, graphene, graphene oxide, carbon nanotube, etc.), and conductive polymers to non-conductive substrates. Metallic particles and carbon-based additives showed outstanding mechanical, electronic, and thermal properties resulting in a wide variety of applications in batteries [50,51], supercapacitors [52], sensors [53], etc. Besides the cost concern of these fillers, additional modifications have to be conducted to such fillers to enhance the dispersity behavior in the polymeric substrates. Moreover, the concentration of such filler has been reported as a very critical parameter for controlling the electrical performance of the final composite. Conductive polymers are considered as a new generation of conductive substrates and show electrical conductivity that is similar to that of other conductive materials.

### 2.1. Electrically Conductive Polymers

Electricity has been linked, in our minds, with wires from a long time ago since electric devices appeared. From those times, it has been thought that metals are the best way to conduct electric current; these metals are frequently found in everyday devices such as bulbs, phones, computers, TVs, etc. Currently, in many applications, non-metal substrates such as conductive polymeric materials that can act as conductors are replacing the metal ones [54]. 

Electrons of good conductive metals are delocalized as the valence (outermost) electrons of a metal atom, which are held loosely, thus allowing them to flow more freely. Polymers can also be as conductive as metals when the monomeric units can form in the polymer in a so called conjugated system [55]. In such a system, the polymeric chain backbone contains alternating double and single bonds [56,57]. This alternating structure enables overlapping p-orbitals in which the π-electrons are loosely bound and accordingly can flow as shown in Figure 1 of poly(acetylene) [58]. 

However, the conjugated polymer has to be doped by the addition of electrons (reduction) or the removal of electrons (oxidation) to promote electrons to flow. In the doped conjugated polymers, the π-electrons can move around the polymer molecules [59,60]. Accordingly, undoped conjugated polymers conduct electricity in a very low semiconductor range 10^−2^–10^−8^ S/cm. In contrast, the electrical conductivity of the doped conjugated polymers increases several fold to meet the requirement of different electric applications such as rechargeable batteries [61,62], solar cells [63,64,65], chemical, and gas sensors [66,67,68], etc. (Figure 2) [53].

In the light of the electrochemical principles that are described above, only a few polymeric structures meet these requirements; the most used are polyaniline [69], polypyrrole [70], polyacetylene [71], polycarbazole [72], polythiophene, and its derivatives [73] (Figure 3). In the following sub-sections, we will focus on conductive polymers which can be potentially synthesized from monomers that are available from renewable resources.

#### 2.1.1. Polyaniline (PANI)

Polyaniline is considered an intrinsically conducting polymer that is synthesized by chemical or electrochemical polymerization of aniline [74]. PANI is regarded as the most interesting of the conductive polymers due to its ease of synthesis from very low-cost monomer, varied properties, and stable electric performance compared to other polymers [75]. Aniline has been known for decades and is derived from fossil raw materials and used for dying cotton fabrics [76]. Very recently, aniline has been derived from biomass instead of petrochemical precursor [77] by the fermentation of sugar [78] and finally commercialized by Covestro.

The electrochemical polymerization of aniline has many advantages over other techniques because it does not need a special procedure for the purification of PANI from solvent, unreacted monomer, and oxidizing agents. It occurs on an electrode that is made of an inert conducting substrate in an aqueous solution of low pH [79]. However, the chemical polymerization of aniline (Figure 4) using various oxidants, most commonly ammonium persulphate, in strong acid medium at 0 °C results in two structural units; reduced phenylenediamine unit and oxidized quinone diimine unit (Figure 4III) [80,81]. The PANI chain is electrically conductive only when the ratio of these two units is 1:1 and is known for its greenish color [82].

#### 2.1.2. Polypyrrole (PPy)

The polymerization of pyrrole [83] can be performed either by chemical oxidation [70] or electrochemically [84]. In the first step, the oxidation is accompanied by an electron release from the pyrrole ring, forming a radical cation [85]. In the next step, the two generated radical cations are coupled, followed by the deprotonation of two hydrogen atoms to yield bipyrrole (Figure 5a). This step is repeated many times to produce the polymer chains. In addition, the radical cation can react with the pyrrole ring to produce a polymer by a chain-growth polymerization mechanism (Figure 5b). Both mechanisms are expected to take place simultaneously. Similarly to PANI, PPy also requires a doping agent to increase the electrical conductivity [86]. Recently, a single-step conversion of renewable furfural to pyrrole in 75% yield was reported, therefore, PPy can be also considered as a potential renewable polymer [87].

#### 2.1.3. Polyacetylene (PA)

Polyacetylene (PA) is constructed from the polymerization of acetylene to obtain polymer chains of repeating units of olefin [88,89]. PA is another polymer which can be considered as potentially renewable, as acetylene can be produced from calcium carbide which is obtained from calcium carbonate [90,91]. PA is considered as the first conductive polymer compared to metals and it exists in two isomeric forms: trans and cis conformations (Figure 6); the highest electrical conductivity is achieved for the trans conformation [92]. 

Initially, PA was polymerized by using Ziegler–Natta catalysts [93,94,95] in which acetylene gas was used. Lately, PA has been produced by using radiation polymerization via ultraviolet [96], gamma [33], or glow-discharge [97] radiations. Recently, PA has been synthesized via ring-opening metathesis polymerization using a cyclooctatetraene (COT), which is much easier to handle than acetylene gas [73]. COT could be isolated from certain fungi [98] or produced from 1,4-butadiene, which is also available from natural resources [99,100,101,102]. In general, the instability towards air and the difficulty in processing make the applications of PA very limited compared to the other conductive polymers.

#### 2.1.4. Polythiophene (PT)

Polythiophene can be produced by oxidative polymerization of thiophene using ferric chloride at ambient temperature (Figure 7). In addition, in a voltaic cell, a PT film can be produced electrochemically on an anode from a solution of thiophene mixed with electrolyte solutions [103].

Doping by using any oxidizing agents, as shown in Figure 8, is a very critical step to obtain an electrically conductive PT salt form. Among different oxidizing agents that are used for doping of PT, highly electrical conductive PTs were achieved by using iodine and bromine [104].

The production of thiophene from furan, which can be prepared from renewable furfural, was also reported. However, it is worth mentioning that the highly toxic hydrogen sulfide is needed in the synthesis [105].

### 2.2. Carbon-Based Electrically Conductive Fillers

Nowadays, carbon nanotubes [106], carbon black [107], graphite [108], and carbon fibers [109] are mostly employed to obtain electrically conductive composites. This is thanks to their chain-like structure and the ability to form electric conductive networks, especially for the carbon fibers [110]. In general, carbon-based fillers are produced from thermal treatments of organic carriers with or without inorganic additives. 

Commonly, the conductivity profile of the carbon-based filler concentrations in the matrix followed the S-shaped (Figure 9) curve, which means that the conductivity can increase dramatically in a narrow loading range of the filler [111].

The critical aspect of incorporating such a carbon-based filler into the composite matrices is that the filler concentration which must be as low as possible to retain the mechanical and physical properties of the hosting matrix. Also, dispersity in the polymeric medium is another challenge to the carbon-based fillers. Therefore, graphene, reduced graphene oxide and graphene derivatives [112] have attained great attention to obtain well-distributed electric conductive fillers [112].

### 2.3. Metallic-Based Media 

Many nanometer to micrometer-sized metallic particles have been used as electrical conducting agents for non-conducting polymers. Silver [113], nickel powder [114], zinc [115], copper [116], and many other agents have been mixed at various concentrations in solid composites or flexible substrates using different techniques for numerous applications, such as anti-static materials [117], sensors [47,118,119], electromagnetic interference shielding material [120], and photovoltaic cells [121]. It is acknowledged that metal salts, such as lithium chloride, are insulators and do not conduct electricity. However, when these metal salts are dissolved in electrolytes (in battery applications), they dissociate to ions that facilitate charge to flow [122]. Here, a confusion can happen by considering metal ions as conductive media. Typically, authors measure the metal salts’ effect on the ionic conductivity by assembling a voltaic cell for measurements and results come by Siemens units.

## 3. Preparation of Electric Conductive Hydrogels Based on Natural Polymers

### 3.1. Irreversible Electrically Conductive Hydrogels 

Irreversible hydrogels are highly interesting due to the ability of irreversible bond dissociation under controlled and stimulated conditions, simple preparation, and stability under various pH values [123]. Irreversible hydrogels of biopolymers are mostly formed by covalent bonds between liner or branched polymers with di/multifunctional cross-linking agents [124]. Accordingly, the mechanical performance, stiffness, and microstructure of this type of hydrogel are precisely tuned by the density of the cross-links.

Irreversible electrically conductive hydrogels that are based on biopolymers, listed in Table 1, have been demonstrated by incorporating electrically conductive media. Such hydrogels can be prepared by mixing of conductive medium with hydrogel precursors followed by the cross-linking step. In addition, in the case of conductive polymers, there is an additional way in which conductive hydrogels can be produced, namely by the synthesis of conductive polymers in pre-formed hydrogels. Given this specificity, here we will report several examples of production and properties of electrically conductive irreversible hydrogels of natural polymers that contain conductive polymers namely PANI, PPy, and PT. It should be pointed out here that a special precaution that is taken during the acetylene gas polymerization [125] makes the fabrication of electrically conductive hydrogel that is based on PA very difficult and, to our best knowledge, there are no data on the fabrication of the conductive hydrogel that is based on PA and natural polymers up-to-date.

The preparation of conductive hydrogels that are based on PANI has shown tremendous attraction towards different applications such as water treatment [69], supercapacitors [126], and bone tissue engineering [7]. Bagheri et.al [69] first prepared hydrogel by the polymerization of acrylic acid in the presence of carboxymethyl cellulose (CMC), a small amount of a cross-linker, and ammonium persulfate (APS) as an initiator (Figure 10). Since the formed sulfate anion radicals can trap the hydrogen from the hydroxyl groups of the CMC, grafting of the acrylic acid onto CMC can occur as well during the polymerization and, thus, it is directly chemically incorporated into the hydrogel structure [21]. The CMC hydrogel was immersed into aniline solution which was polymerized by ammonium persulfate and doped by hydrochloric acid. The prepared conductive hydrogel showed an electrical conductivity as high as 0.75 S/cm. The conductivity was similar to the conductivity value of 0.65 S/cm that was determined for the PANI-based hydrogel that was prepared purely from polyacrylamide [30].

Conductive acrylamide-grafted CMC hydrogel was reported by Suganya et al. [126], with the same synthetic approach as in the case where polyacrylic acid-grafted CMC hydrogels was used. The authors, however, used a five times lower concentration of aniline during the oxidative polymerization. Therefore, the electrical conductivity of the prepared hydrogel was significantly lower and reached a value of 2.71 × 10^−4^ S/cm.

The CMC hydrogel was prepared also using glycerol diglycidyl ether (GDE) as a cross-linker in an alkaline medium at 40 °C for 24 h [127]. Similarly, as in previous cases, the hydrogel was immersed into the aniline solution and subsequently polymerized by ammonium persulfate, while in this case it was doped by benzene sulfonate. The results showed that increasing electrical conductivity was obtained by decreasing the CMC and/or GDE concentration. The doping with benzene sulfonate increased the electrical conductivity, reaching the maximal value of 6.31 × 10^−3^ S/cm. 

Khorshidi et al. [7] used oxidized polysaccharides such as oxidized alginate and hyaluronic acid that were mixed with gelatin and conductive filler, such as graphene. The mixture was spontaneously gelled through a Schiff-base mechanism in the presence of electrospun fibers that were prepared from a solution of PANI and polycaprolactone (PCL) (Figure 11). Due to the small portion of PANI, the final composite hydrogel showed quite low electrical conductivity of 10 ± 1 × 10^−6^ S/cm. That was, however, sufficient for the application as tissue engineering scaffolds with improved adhesion, spreading, and proliferation of osteoblast-like cells.

An electrically conductive cellulose-based hydrogel containing PANI was prepared by Xu et al. [128]. The regenerated cellulose hydrogel was prepared by casting a high concentration solution of sodium hydroxide and urea (12%). The authors invented an apparatus in a U-shape to conduct interfacial polymerization of aniline on one side of the cellulose hydrogel. Meanwhile, PANI has been obtained by the oxidation of aniline via ammonium persulfate and doped by a self-cross-linking agent, phytic acid, as shown in Figure 12. The obtained scaffold showed an electrical conductivity as high as 0.49 S/cm.

Electrically conductive hydrogels that are based on PPy have been investigated for different applications such as sensors [133], medical purposes [32], flexible supercapacitor electrodes [134], and electronic devices [135]. The common route to obtain PPy was via conducting the polymerization step in the presence of either the hydrogel or the hydrogel precursor. Yang et al. [32] prepared an irreversible conductive hydrogel that was based on HA using PPy for a dual effect, i.e., as a cross-linker and ECP. The authors first coupled 3-aminopropylpyrrole to HA chains to obtain pyrrole groups that were attached to the polymer chains (Figure 13). Then, pyrrole monomers were copolymerized with the coupled ones to produce a propylpyrrole-cross-linked hydrogel. Such a hydrogel was soft (~3 KPa) and showed electrical conductivity of ~7.3 mS/cm.

An electrically conductive composite hydrogel was prepared using nanocrystalline cellulose that was grafted by acrylic acid in the presence of a cross-linker and APS [129]. The oxidative polymerization of pyrrole was conducted in the presence of the grafted-nanocellulose. PPy was doped by sodium p-toluenesulfonate to provide electrical conductivity up to 8.8 × 10^−3^ S/cm. The hydrogel composite was very stable and with the significantly increased compressive modulus of 4.16 MPa compared to 0.23 MPa for pure hydrogel, while still achieving a water retention capacity as high as 910%. 

Chemically cross-linked chitosan was also used to prepare electrical conductive hydrogels by graft-polymerization of acrylic acid [130]. The fabricated hydrogel was stirred with PPy, doped with ferric chloride in the presence of polyethylene glycol for better homogenization, and subsequently mixed also with magnetite nanoparticles (Fe_3_O_4_). The composites with electric conductivity up to 10^−3^ S/cm were prepared in this approach. Since the hydrogel was stirred with the additives, such as PPy and Fe_3_O_4_, the final composite was not in the form of a compact hydrogel. Thus, this synthetic approach can be used only for limited applications.

Contrary to previous work where the compact chemically cross-linked hydrogel was destroyed during stirring with PPy, Kashi et al. [131] fabricated an injectable hydrogel that was based on physically cross-linked chitosan using β-glycerophosphate that was mixed with PPy oligomers. The authors prepared PPy in an imidazolium-based ionic liquid that was oxidized by ammonium persulfate and doped by sodium perchlorate. The electroactive hydrogel with electrical conductivity in the range of 1.9–4.4 × 10^−3^ S/cm, depending on the PPy content, was prepared. This hydrogel was made for cartilage tissue engineering to promote tissue repair and regeneration. The soft electrically conductive hydrogels provide a new level of control over biomaterials that are applied into the human body, especially in nervous and cardiac tissue engineering where conducting electricity is a key of successful function [136]. The conductivity values that were achieved in the work of Kashi et al. were still about one order lower than the normal electrical conductivity of cartilage tissue. However, higher oligopyrrole content was not tested to avoid cytotoxicity of the scaffolds.

Chitosan and chitosan derivatives can be mixed with other natural polymers in the presence of PPy to obtain a chitosan-based conductive hydrogel [132]. Thus carboxymethyl chitosan was mixed with alginate solution and PPy that was pre-synthesized by oxidation polymerization using ammonium persulfate and doped by hydrochloric acid. The mixture was then physically cross-linked via calcium cation by using CaCO_3_ and _D_-glucono-δ-lactone to obtain the electrically conductive hydrogels. Such conductive chitosan-based hydrogel, fabricated for peripheral nerve regeneration, showed an electrical conductivity in the range of 2.41 × 10^−5^–8.03 × 10^−3^ S/cm depending on the PPy loading. 

Unlike PANI and PPy, a few studies [11,73] have described the synthesis of electrically conductive hydrogels that are based on natural polymers and PTs. Pairatwachapun et al. [11] fabricated an electrical conductive hydrogel that was based on carrageenan and PT to fabricate a transdermal patch for delivery of topical drug, namely acetylsalicylic acid (ASA). The authors used physical cross-linkers (CaCl_2_, MgCl_2_, and BaCl_2_) to fabricate a hydrogel in the presence of ASA. PT was synthesized separately by an oxidative polymerization using ferric chloride and added to the carrageenan hydrogel precursor before casting. The fabricated matrix was utilized for the electric field-assisted drug delivery which drastically enhanced the drug delivery rate [11]. Unfortunately, no data for the electrical conductivity measurement was mentioned for this conductive hydrogel. Another research group, Pattavarakorn et al. [73], fabricated an electrically conductive hydrogel that was based on carboxymethyl chitosan/chitosan/PT by using glutaraldehyde as a cross-linking agent. PT was polymerized in toluene separately and mixed with the polymeric mixture followed by the chemical cross-linking. The electroactive performance was tested by measuring the bending response under an applied electric field while the conductive data were not measured.

### 3.2. Self-Healing Electrically Conductive Hydrogels 

A self-healing hydrogel means that the hydrogel matrix is capable of self-repair after any mechanical damage, rupture, crack or fracture. The mechanism varies depending on the chemical structure of the polymeric backbone as well as the type and chemical structure of the cross-linking points, which can be sensitive to various healing conditions, such as a specific temperature, pressure, or the exposure of a particular type of radiation [29]. The self-healing phenomenon can occur based on two different approaches, namely extrinsic and intrinsic [137]. The extrinsic approach is based on one time healing when external pressure causes cracks in the substrate that contains microcapsules that are loaded with healing agents (cyanoacrylate, thiol-acrylate, epoxy, etc.) that can instantly react and heal the cracks and ultimately produces a curable matrix [138]. Subsequent healing of the mechanical damage in the same position is, however, limited. On the other hand, the intrinsic approach relies on dynamic reversible bonds via either physical or chemical bonds, and, therefore, it is the most applicable approach for many applications [119,139,140].

The original thought of self-healing substrate was the thermoplastic polymer that converts to liquid at high temperature and turns back to solid at cold temperature. In hydrogels, there are two main mechanisms that can lead to self-healing behavior based on the dynamic covalent bonds in chemical cross-links or non-covalent bonds in physical cross-links (Figure 14) [141].

Dynamic covalent bonds can act as self-healing agents in hydrogels when bonds such as carbon/nitrogen bonds (hydrazone and imine chemistries), boron-ester bonds, and disulfide bonds, or bonds that are formed based on reversible radical or Diels–Alder reactions are present. 

Meanwhile, non-covalent bonds, i.e., physical cross-links, can also show a great potential to provide self-healing hydrogels via multiple hydrogen bonding interactions, ion interaction, metal-ligand interaction [142], host/guest interaction, polymer/nanocomposites interactions, or hydrophobic interactions [139].

So far, there are only a few works describing the preparation and utilization of self-healing electrically conductive hydrogels filled with any conductive media. A summary of such hydrogels that are based on both chemical and physical cross-linking can be found in Table 2. In addition, all these systems are also discussed in the following paragraphs. For the self-healing systems which were not so far used for the preparation of electrically conductive hydrogels that are based on natural polymers. Here, examples of self-healing systems that are applied for non-conductive hydrogels that are based on natural polymers and/or examples of self-healing systems that are applied for conductive hydrogels that are based on synthetic polymers are shown to demonstrate the possible applicability of such self-healing systems for electrically conductive hydrogels of natural polymers.

#### 3.2.1. Chemically Cross-Linked Self-Healing Electrically Conductive Hydrogels

Among the reversible chemical reactions, a great interest is focused on the ones which can provide self-healing of the hydrogels under mild conditions, such as room temperature, with at least 90% efficiency of the mechanical damage repair.

***Imine bond*** is a very dynamic chemical reaction between amino groups and aldehyde functional groups (Figure 15) unless a reduction of the imine occurs as an adverse reaction.

Ren et al. [143] fabricated a self-healing electrically conductive hydrogel that was based on aminated gelatin, dialdehyde alginate, and PPy (Figure 16). PPy was polymerized via an oxidizing agent, ammonium persulfate (APS) in the mixture of the gelatin and alginate derivatives at −20 °C. Subsequently, the solution temperature was raised to allow the formation of imine bonds and to create the electrically conductive hydrogel. The mechanical strength of the obtained hydrogel reached 0.5 MPa. The electrical conductivity that was recalculated from the resistance reached 1.4 × 10^−5^ S/cm. The authors demonstrated the application of the hydrogel to serve as repairable wires enabling lightening of the LED bulbs, while complete self-healing of two separate parts occurred in 40 min after their connection. In addition, since the hydrogel possessed good flexibility, the authors showed the dependence of resistance on the angle of bending or change length under compression proving a potential application in soft sensors and biocompatible devices. It is worth pointing out, as a suggestion for future systems, that more stable hydrazone groups can be formed between hydrazine and aldehyde/ketone, while the hydrogel can be achieved faster compared to the hydrogels that are based on imine bonds [148].

***Polyol/borax*** self-healing hydrogels showed superior advantages over the rest of the covalent bond mechanisms as the hydrogel can be formed without any need for chemical modifications of the polymeric components. Figure 17 shows the simple mechanism of borax dissociation in water into boric acid and borate ions that can chemically cross-link the hydroxyl-containing polymers, such as carbohydrates and polyols, through boron ester bonds. An alkaline medium is preferable for this reaction to provide densely cross-linking hydrogels as more borate ions (B(OH)_4_^−^) are available over the boric acid (B(OH)_3_) [140].

In the practical application of this phenomenon, self-healing, conductive, and adhesive hydrogels have been fabricated based on sodium hyaluronate chains. The authors [122] first esterified hyaluronic acid with dopamine to prepare it for a reaction with borax. The cross-linking with borax was performed in one step with in situ free radical polymerization of acrylamide in the presence of a diacrylamide cross-linker. Thus, in addition to the polyol/borax covalent bonds, hydrogen bonds between the hydroxyl groups of catechol units and amide groups of polyacrylamide were also present in the final hydrogel (Figure 18). Residual-free hydroxyl groups of catechol also provided self-adhesion properties of the hydrogel. The obtained hydrogel showed a high toughness character of 42.4 kPa and real-time for healing was 1 h. In this work, lithium chloride salt was used to increase the electrical conductivity up to 1.1 × 10^−2^ S/cm compared to 1.8 × 10^−4^ S/cm that was determined for the salt-free hydrogel. It should be pointed out here that the authors used phosphate buffer silane in the hydrogel preparation and that might be the reason for the relatively high conductivity that was determined for salt-free hydrogel samples.

***Diels−Alder reaction*** is considered a promising strategy to obtain self-healing hydrogels that are promoted by a repeated healing character in a wide range of temperatures. In general, it is a [4 + 2] cycloaddition reaction between dienes and dienophiles under the formation of cyclohexene derivatives as shown in Figure 19 [149].

The self-healing behavior that is based on the Diels-Alder reaction has been demonstrated for natural polymers such as the chemistry of fulvene-modified dextran/dichloromaleic acid-modified poly(ethylene glycol) [150], furyl-modified cellulose nano-crystal/maleimide-end-functionalized PEG [151], and furan-modified pectin/maleimide-modified chitosan [152]. However, to the best of our knowledge, the Diels-Alder reaction-based self-healing has not been published so far to develop conductive hydrogels from natural polymers. As an example that Diels Alder reactions can be successfully used in the presence of conductive media, Lin et al. prepared self-healing hydrogels that were filled with graphene oxide and silver nanowires that were based on polyurethane. The electrical conductivity of this hydrogel was in the order of 10^−3^ S/cm. The cross-linking was performed based on the maleimide terminal functional groups that reacted with the furan dangling functional groups from the polyurethane backbone [153]. In a similar way, both the furan and maleimide groups could also be bound to natural polymers with the aim to produce self-healing conductive hydrogels.

***Disulfide bond*** strategy has also been employed to obtain self-healing hydrogels due to the advantages that include its relatively high bond energy (251 kJ/mol), which leads to strong bonds between molecules and, at the same time, their reversible reactions at low temperatures that enable self-healing behavior under mild conditions [139]. The disulfide S-S-bond is formed by the coupling of two thiol groups during the oxidation process, as depicted in Figure 20.

As self-healing electrically conductive hydrogels that were based on natural polymers were not reported so far, only an example of self-healing non-conductive hydrogel that was based on natural polymers containing disulfide bonds can be presented here to prove the potential of this type of reversible bond to be applied for the preparation of conductive systems as well. Thus, Shu et al. [154] fabricated a self-healing hydrogel based on disulfide chemistry using hyaluronic acid as a polymer. First, the authors synthesized dithiobis(propanoic dihydrazide) (DTP) and dithiobis(butyric dihydrazide) (DTB). These two thiol-containing compounds were linked to HA via a coupling agent (carbodiimide) to achieve dangling thiol groups along the HA chains (Figure 21). Hydrogels were formed at pH values from 7 to 9 with rapid gelation behavior. However, no data were mentioned about the real-time for either the healing or the mechanical strength.

As an example that self-healing based on disulfide bonds can be successfully used in the presence of conductive media, the electrically conductive self-healing hydrogels that were based on polyurethane were reported by Zhanyu et al. [155]. In that work, diisocyanate-terminated urethane prepolymer was synthesized based on polyethylene glycol (PEG, M_n_ = 2000 g/mol) to react with 4-aminophenyl disulfide (dithiodianiline) to obtain the self-healing hydrogel. The electrically conductive hydrogel was obtained when the lyophilized hydrogel was allowed to reswell in various concentrations of pyrrole/isopropanol solution. In situ polymerization of pyrrole was conducted in the presence of ferric nitrate. The obtained hydrogel combined multifunctionality with an electrical conductivity of 5.5 × 10^−4^ S/cm, moderate tensile strength of 1.1 MPa, and a self-healing real-time of 10 min.

#### 3.2.2. Physically Cross-Linked Self-Healing Electrically Conductive Hydrogels

***Hydrogen bonding*** of super-macromolecules, inspired by biomolecules, has an important role to obtain a cross-linking network, which exhibits reversible interactions, providing a self-healing character. Agarose belongs to the linear polysaccharides that gel reversibly in water by a change of temperature. Hur et al. [144] polymerized PPy in agarose warm solutions using copper chloride as an oxidizing agent to obtain self-healing electrically conductive hydrogels upon cooling down. The final matrix showed an electrically conductive value of 0.35 S/cm. 

An additional self-healing system that is based on multiple hydrogen bonds that are formed between carboxyl, hydroxyl, amino, and acetamide groups was reported by Cao et al. [119], who prepared a self-healing electrically conductive hydrogel from carboxyl cellulose nanocrystals in combination with chitosan-based decorated epoxy natural rubber latex. The electrically conductive behavior was gained from the carbon nanotubes that were embedded in the hydrogel matrix. The obtained hydrogel exhibited a real-time self-healing capability within only 15 s. The mechanical strength of the healed samples showed the same values as the original samples (0.8 MPa). The hydrogel electrical conductivity was only in the order of 10^−8^ S/cm, which was, however, inferred as sufficient for application as sensors for human–machine interactions. 

***The ion interactions*** mechanism is another non-covalent approach to achieve self-healing hydrogels in which metal ions in specific oxidation levels coordinate with lone pairs of electrons on polymeric chains. Darabi et al. [145] utilized this approach to obtain an electrically conductive self-healing hydrogel that was based on chitosan and PPy. In the first step, PPy was grafted to the pendant double bond of the pre-functionalized chitosan. In the second step, acrylic acid was graft-polymerized and cross-linked the chitosan-PPy chains in the presence of iron III (Fe^3+^) to achieve a double network of chemical and physical cross-linking. The self-healing property was then based on the reversible ionic interactions between the ferric ions and the carboxylic groups of PAA and/or NH groups of PPy (Figure 22). The real-time to reach 100% of original mechanical strength recovery was only 2 min. The obtained hydrogel showed an electrical conductivity as high as 5–10 × 10^−2^ S/cm.

***Hydrophobic interactions*** are also capable of providing instant self-healing hydrogels without any external stimulus with an easy preparation methodology. Yang et al. [146] investigated the combination of multiwalled carbon nanotubes (MWCNTs) with polyacrylamide hydrogel by the utilization of cellulose nanofibers (CNF) as a dispersant. The self-healing character is provided by the hydrophobic interaction between the CNF and polyacrylamide chains. MWCNTs provided good electrical conductivity of 8.5 × 10^−3^ S/m while the CNF supported the mechanical strength to reach 0.24 MPa. The real-time self-healing was 10 min to recover the original shape. 

***Host-guest interaction*** can be used to obtain self-healing substrates based on the hydrophobic interactions between the two moieties as well. Liu et al. [147] fabricated a self-healing hydrogel based on regenerated silk fibroin (SF) substrate. The authors introduced beta-cyclodextrin (β-CD) molecules to the SF backbone by a reaction of monoaldehyde β-CD with amino groups that were present in the SF structure. The mechanism of self-healing relies on the host-guest interactions between β-cyclodextrin and the aromatic groups of the amino acid side chains of SF, such as tyrosine, tryptophan, phenylalanine, and histidine. To increase the host-guest interactions, ethynylbenzene groups were also attached to the SF backbone using an azo bridge. The electrically conductive behavior was introduced by the in situ polymerization of pyrrole using APS as an oxidant and laccase as a catalyst. In addition, chemical cross-linking was introduced photochemically forming dityrosine cross-links. Figure 23 shows the proposed mechanism of self-healing of the matrix.

The obtained hydrogel showed an electrochemical conductivity of 1 × 10^−3^ S/cm. Meanwhile, the dissociation force was 4.4 × 10^−3^ N and 1.1 × 10^−2^ N for one piece and two pieces of the hydrogel, respectively. The healing mechanism relied on pressure to repair the damage in the hydrogel but the real-time healing was not reported.

## 4. Conclusions & Future Prospects

Electrically conductive substrates hold a prestigious position, especially in the map of electrical applications. Increasing the surface area of the main components of conductive substrates showed a significant increase in the electrochemical performance of such substrates. The electrospinning technique, to fabricate nano-sized fibers, can be utilized in the coating, decorating, or constructing of electrical components, which are considered as a turning point to overcome the drawbacks of the traditional way of fabrication. Generally, electrospun fibers are made of polymers that are capable of being carbonized by thermal treatments to provide carbon nanofibers or nanotubes. Such carbon segments, when mixed with different substrates, enhance electro-conductivity. Also, nanoparticles of inorganic metals and heteroatom-doped metals enhance the electro-conductivity of any hosted substrate. Impregnated polymers with such electro-conductive segments have been electrospun either in the form of single filaments, core/shell structures, or hollow fibers. The electrospun carbon nanofibers or nanotubes have a very large surface area that facilitates the electron transfer in a much higher magnitude compared to regular composites [156,157]. Also, heteroatoms-doped carbon materials that are based on electrically conductive polymers can provide electrocatalysis active sites to the matrices for various applications i.e., excellent oxygen reduction reaction [158]. 

Self-healing mechanisms are classified into non-covalent and covalent bonds by which the different matrices have been fabricated. The combination of such electrically conductive polymers and hydrogels that are fabricated through these mechanisms is yet to be fully investigated. The potential of many biopolymer substrates has not been explored for the fabrication of self-healing matrices. Accordingly, novel materials for many applications can be discovered out of these new combinations. For further development on the utilization of electro-conductive self-healing hydrogels, here are some suggested prospects.
Providing electro-conductive hydrogels in different formulations such as microspheres, electrospun fibers, and adhesive membranes. Preparing such electro-conductive self-healing hydrogels in the form of porous platforms. This will increase the surface area of the substrates and can increase their electrochemical performance several-fold. Imparting an adhesive character to such self-healing electrically conductive substrates will open another application in transdermal drug delivery.


## Figures and Tables

**Figure 1 ijms-23-00842-f001:**
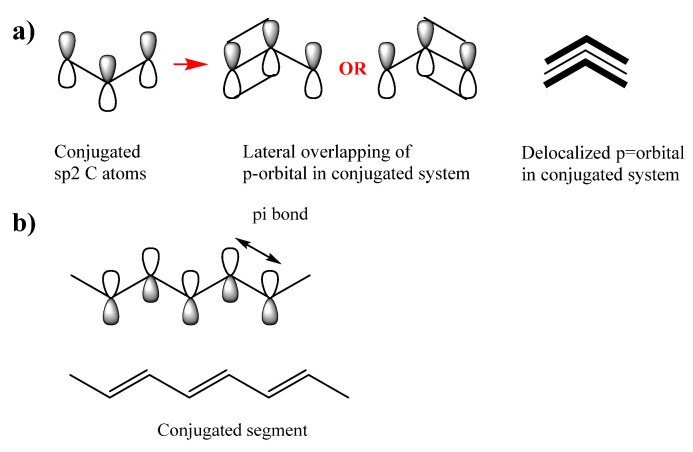
(**a**) Conjugated system of p-orbitals and (**b**) alternating double and single bonds enabling p-orbitals overlapping in poly(acetylene).

**Figure 2 ijms-23-00842-f002:**
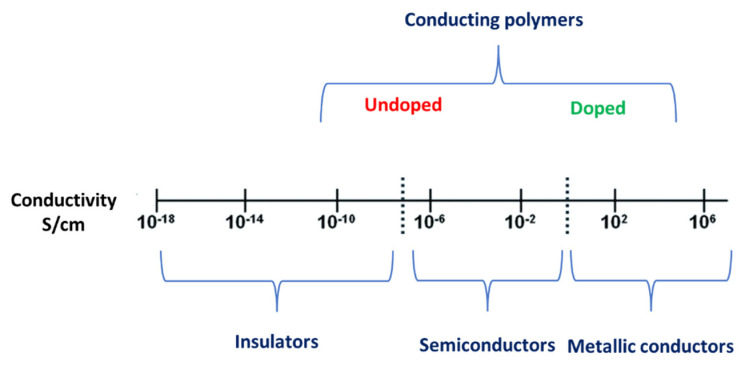
The electric conductivity of materials in S/cm unit.

**Figure 3 ijms-23-00842-f003:**
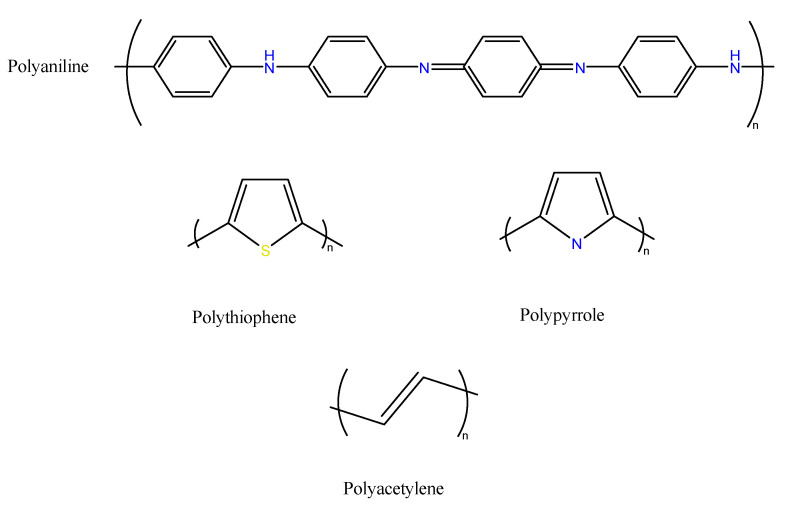
The chemical structures of different conductive polymers.

**Figure 4 ijms-23-00842-f004:**
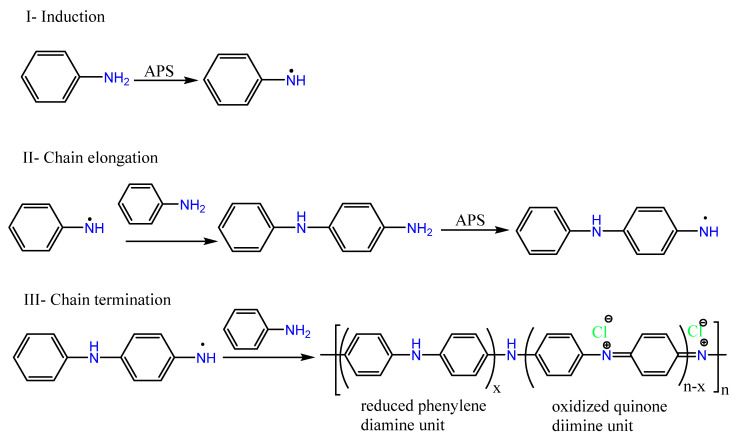
Oxidation polymerization of aniline.

**Figure 5 ijms-23-00842-f005:**
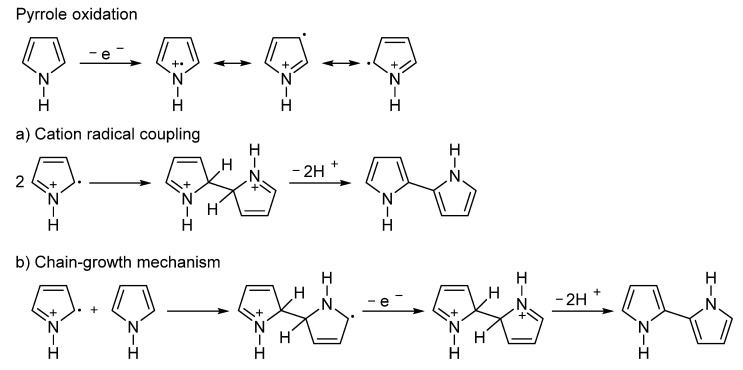
The polymerization of pyrrole via (**a**) cation radical coupling and (**b**) chain-growth mechanism.

**Figure 6 ijms-23-00842-f006:**
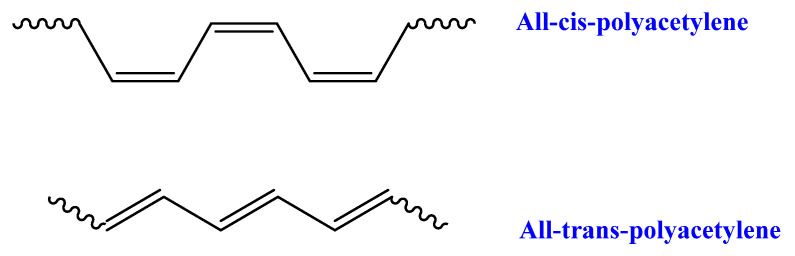
Cis/trans isomers of polyacetylene.

**Figure 7 ijms-23-00842-f007:**
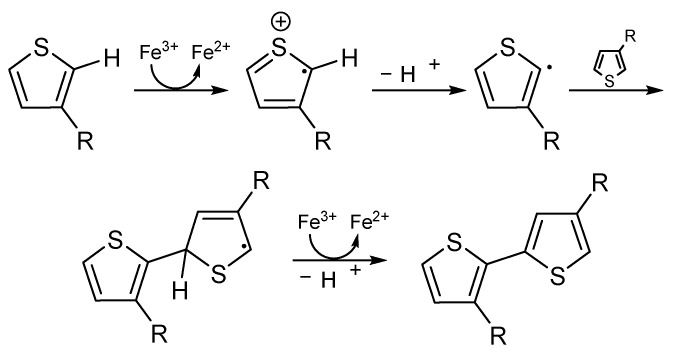
Proposed mechanism of thiophene oxidative-polymerization via ferric chloride.

**Figure 8 ijms-23-00842-f008:**
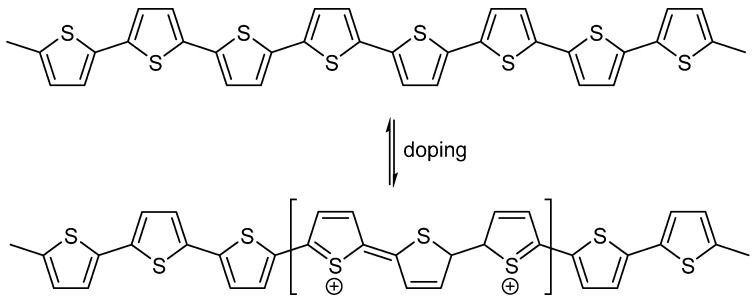
Proposed chemical structure of the electrically conductive PT salt.

**Figure 9 ijms-23-00842-f009:**
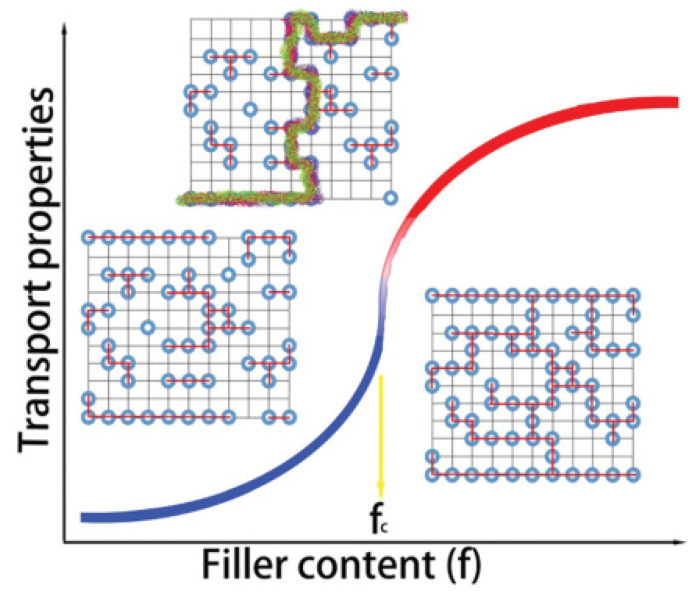
S-shaped curve for the effective electrical conductivity. Reproduced from Ref. [111] with permission from the Royal Society of Chemistry.

**Figure 10 ijms-23-00842-f010:**
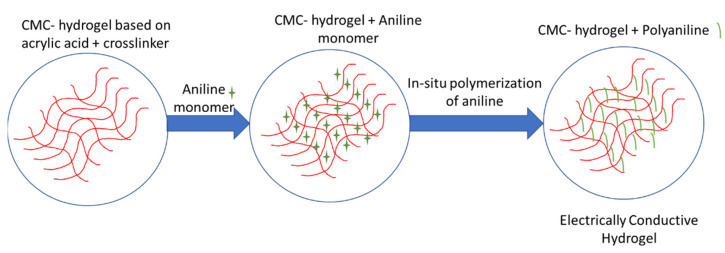
Preparation of carboxymethyl cellulose (CMC)/ polyaniline (PANI) conductive hydrogel.

**Figure 11 ijms-23-00842-f011:**
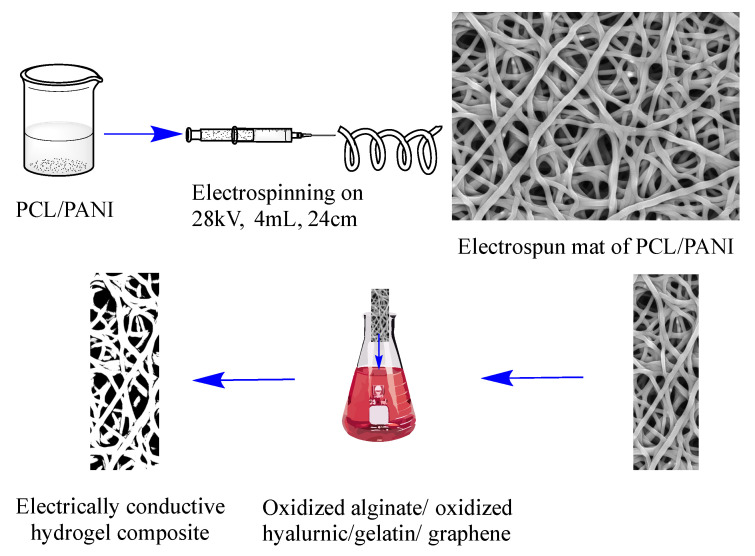
Hydrogel/fiber conductive scaffold that is based on PANI/PCL electrospun fiber and an oxidized polysaccharide/gelatin/graphene composite.

**Figure 12 ijms-23-00842-f012:**
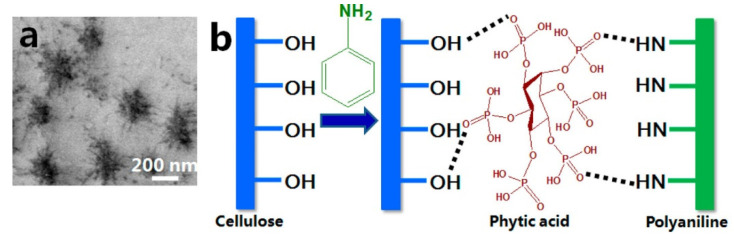
Electrically conductive scaffold that is based on regenerated cellulose and PANI: (**a**) Transmission electron microscope image of the cross-section of the conductive side of PANI/ regenerated cellulose; (**b**) The proposed mechanism of the polymerization reaction and the hydrogen bonding between cellulose/aniline/phytic acid. Reprinted with permission from reference [128].

**Figure 13 ijms-23-00842-f013:**
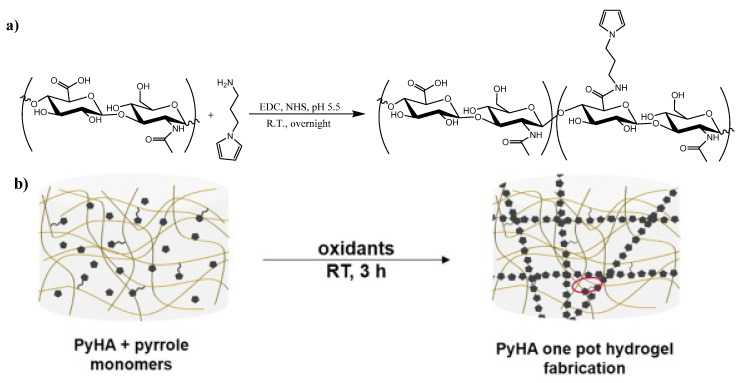
(**a**) Scheme of hyaluronic acid (HA) modification by pendant pyrrole moieties to produce PyHA and (**b**) schematic drawing of the oxidative polymerization of pyrrole in the presence of PyHA. Reprinted with permission from reference [32].

**Figure 14 ijms-23-00842-f014:**
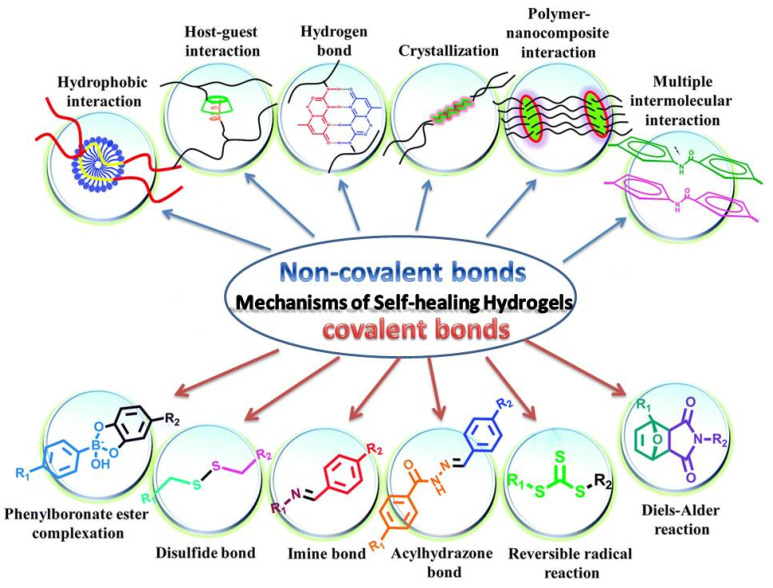
The mechanisms of self-healing hydrogels. Reprinted with permission from reference [141].

**Figure 15 ijms-23-00842-f015:**
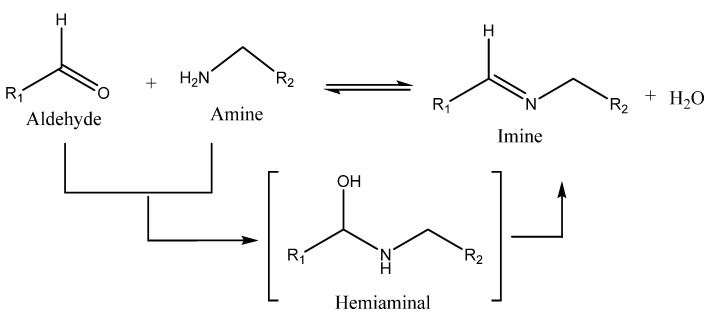
The chemical reaction to imine formation.

**Figure 16 ijms-23-00842-f016:**
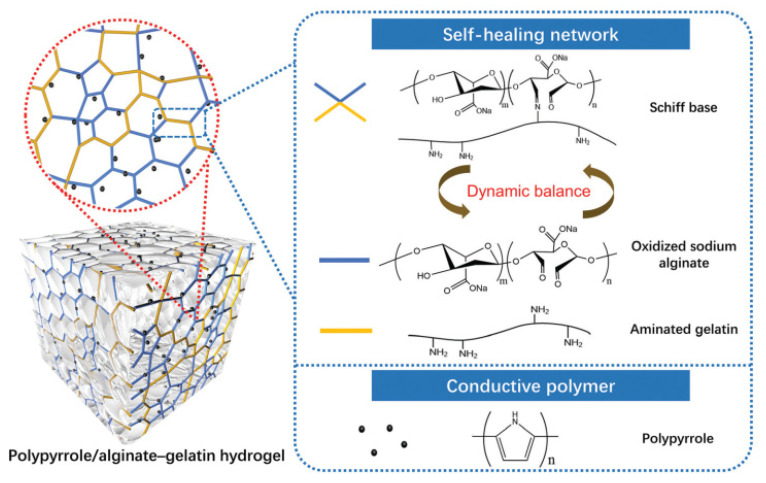
Systematic diagram of a self-healing conductive hydrogel that was based on imine bonds/PPy. Reprinted with permission from Ref. [143].

**Figure 17 ijms-23-00842-f017:**
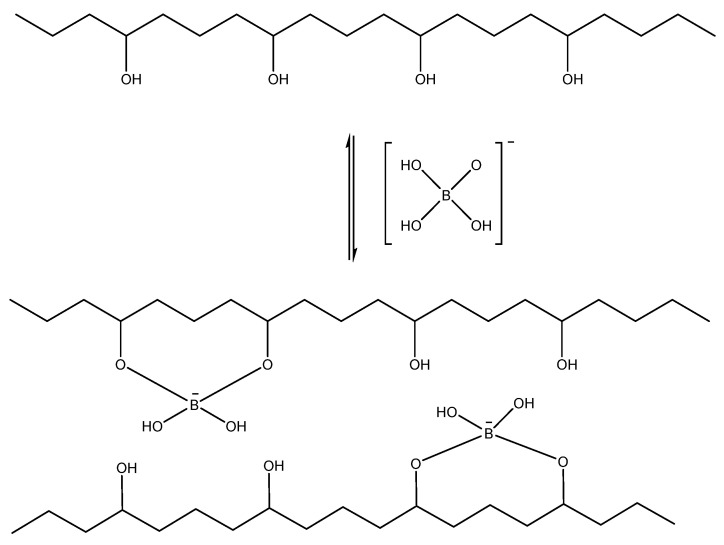
Borax dissociation in water and its cross-linking with polyols.

**Figure 18 ijms-23-00842-f018:**
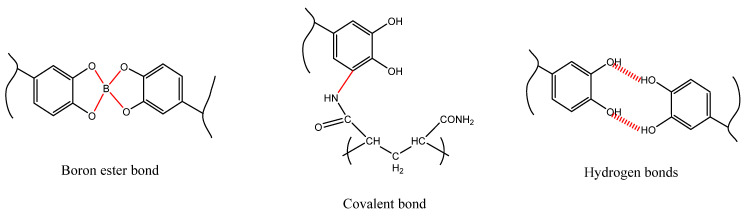
Self-healing bonds in dopamine-modified hyaluronic acid-based hydrogel reported in Ref. [122].

**Figure 19 ijms-23-00842-f019:**
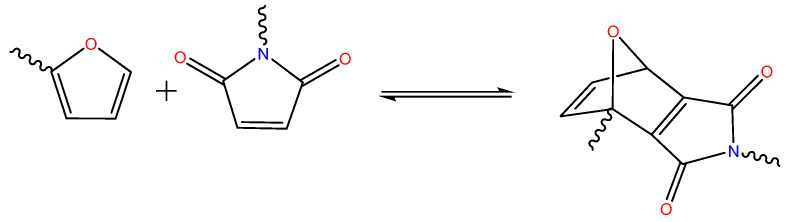
Principle of self-healing based on reversible Diels-Alder reaction.

**Figure 20 ijms-23-00842-f020:**
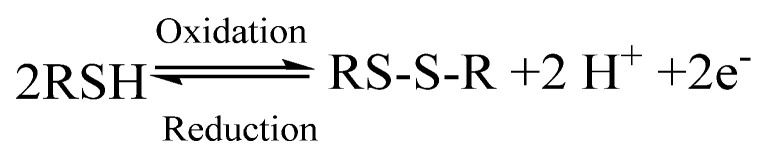
The formation of a disulfide bond through the coupling of two thiols.

**Figure 21 ijms-23-00842-f021:**
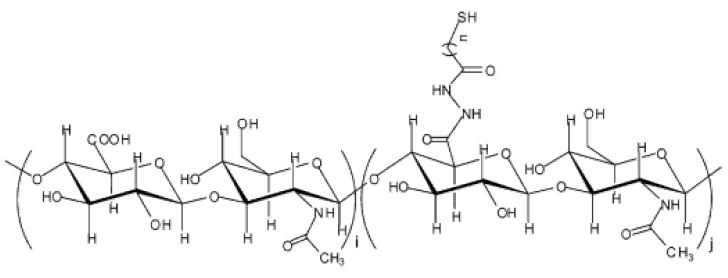
The chemical structure of thiolated-HA; n = 2, HA-DTPH; n = 3, HA-DTBH. Reproduced with permission of Ref. [154].

**Figure 22 ijms-23-00842-f022:**
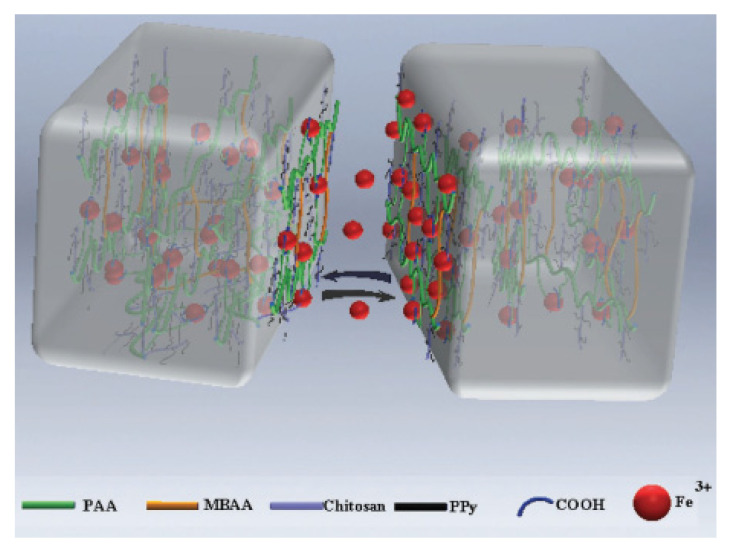
Schematic diagram of the electrically conductive self-healing hydrogel that was based on chitosan, polyacrylic acid, PPy, and ferric ions. Reproduced with permission from Ref. [145].

**Figure 23 ijms-23-00842-f023:**
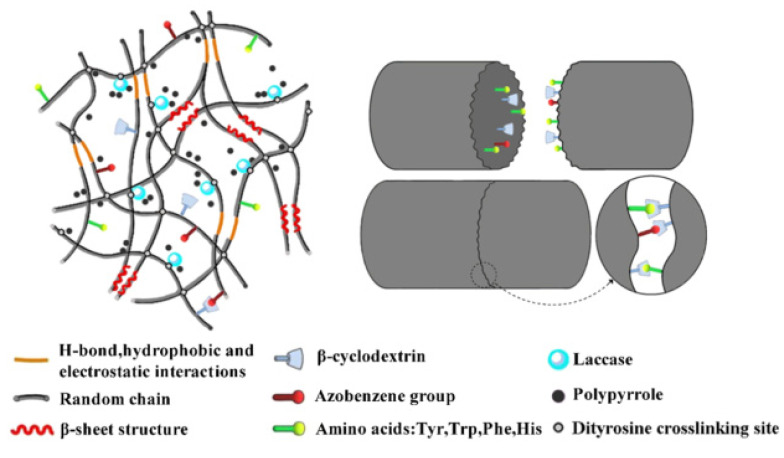
Schematic diagram of the self-healing mechanism of the silk-fibroin-based hydrogel. Reproduced with permission of Ref. [147].

**Table 1 ijms-23-00842-t001:** The composition and properties of irreversible electrically conductive hydrogels that are based on natural polymers substrates that are filled with conductive polymers.

Gelation Mechanism	Polymeric Substrates	Electric Conductive Mediums	Conductivity	Mechanical Performance	References
Graft-polymerization via acrylic acid	Carboxymethyl cellulose	PANI	0.75 S/cm	N/A	[69]
Graft-polymerization via acrylamide	Carboxymethyl cellulose	PANI	2.71 × 10^−4^ S/cm	N/A	[126]
Chemical cross-linking via glycerol diglycidyl ether	Carboxymethyl cellulose	PANI	6.31 × 10^−3^ S/cm	N/A	[127]
Hydrogen bonding interaction via phytic acid	Regenerated cellulose	PANI	2.5 × 10^−2^–6.8 × 10^−1^ S/cm	1.08–2.71 MPa	[128]
Chemical cross-linking via PPy grafted onto HA and polymerized afterward	HA	PPy	~7.3 × 10^−3^ S/cm	3 Kpa	[32]
Chemical cross-linking via acrylic acid and APS	Nanocrystalline cellulose	PPy	8.8 × 10^−3^ S/cm	4.16 Mpa	[129]
Graft-polymerization via acrylic acid	Chitosan	PPy	10^−3^ S/cm	N/A	[130]
Hydrogen bonding interaction via β-glycerophosphate	Chitosan	PPy	1.9–4.4 × 10^−3^ S/cm	N/A	[131]
Ion-interaction via calcium cations	Carboxymethyl chitosan/alginate	PPy	2.41 × 10^−5^–8.03 × 10^−3^ S/cm	N/A	[132]
Physical cross-linkers via CaCl_2_, MgCl_2_ and BaCl_2_	Carrageenan	PT	N/A	N/A	[11]
Chemical cross-linking via glutaraldehyde	Carboxymethyl chitosan/chitosan	PT	N/A	N/A	[73]

N/A: Not available; APS: ammonium persulphate; MPa: MegaPascal; KPa: kilopascal; S/cm: Siemens Per Centimeter.

**Table 2 ijms-23-00842-t002:** Self-healing electrically conductive hydrogels that are based on different healing mechanisms, different substrates, conductivity, and mechanical performance.

	Self-Healing Mechanism	Polymeric Substrate	Electrical Conductive Mediums	Self-Healing Real-Time	Conductivity	Mechanical Performance	References
**Chemical Crosslinking**	Imine bond	Aminated gelatin/Dialdehyde alginate	PPy	40 min	1.4 × 10^−5^ S/cm	0.5 MPa	[143]
	Polyol/borax	Dopamine-hyaluronic	Lithium chloride	1 h	1.1 × 10^−2^ S/cm	42.4 kPa	[122]
**Physical Crosslinking**	Hydrogen bonding	Agarose	Copper chloride	N/A	0.35 S/cm		[144]
		Carboxyl cellulose nanocrystal	Carbon nanotube	15 s	10^−8^ S/cm	0.8 MPa	[119]
	The ion interactions mechanism	Chitosan	PPy	2 min	5–10 × 10^−2^ S/cm	~10 KPa	[145]
	Hydrophobic interaction	Cellulose	Multiwall carbon nanotube	10 min	8.5 × 10^−3^ S/cm	0.24 MPa	[146]
	Host-guest interaction	Cyclodextrin-modified silk fibroin	PPy	N/A	1 × 10^−3^ S/cm	4.4 ×10^−3^ N	[147]

N/A: Not available.

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
