# Peer review of "Irreversible and Self-Healing Electrically Conductive Hydrogels Made of Bio-Based Polymers"

_ijms, 2022, doi:10.3390/ijms23020842_

Round 1

Reviewer 1 Report

This is a well organized paper reviewing a timely topic, the synthesis of self healing electrically conductive hydrogels.  These are seeing many new emerging applications, especially in tissue engineering.

Comments and Edits.

Page 2  Line 71  It would help the reader to add a brief paragraph explaining the statement  “healing process unlike other self-healing mechanisms”.  Could the author expand on this?

P 8  Ln 237   “Then, a confusion can happen…”   This sentence is not clear, could the authors expand on this, as to what they mean.

P 11 Ln 337  “Contrary to previous work”.  Could the authors, in this paragraph make a few comments as to why electrical conductivity is important for some tissue engineering applications?

General Edits

P1  ln 36   insert “are” before “made”       systems, contain

P1 ln 39 delete molecular, insert dry

P1 ln 44     delete its, insert their

P2

Ln 45 delete has been, insert   emerged “a” few

Ln 47      change Biopolymer to Biopolymers

Ln 53       change hydrogels to hydrogel     open to opens              insert “that” spans

Ln 58        change is to are

Ln 73     insert “that” spans

P3

Ln 100    delete “an”

Ln 112   delete “a”

Ln 117   conduc”t”ive   add t

P4

Ln 135   only “a” few  add a

Ln 136   mostly  to “most”

Ln 143    change “as” to an

Ln 152        insert    purification “of” PANI

P5

Ln 170    insert  requires “a” doping

P6

Ln 177 replace “perform” with “obtain”

Ln 206   delete “a”

P7

Ln 226   delete “been”        and great attention”, in order” to obtain

P8

Ln 251  insert there is “an” additional

P9

Ln 276   replace while with “with”

Ln 284   should read “increasing electrical conductivity was obtained” by

Ln 301   “from” should be “on”

P12

Ln 399  should read “So far there are only a few”

Ln 406  replace  “is” with are

Ln 407   replace “for” with of

Ln 413  Imine bond is “a” very

P14

Ln 433     to point “out” here

Ln 449- ln 450   should read “ prepare a modified polyol for reaction with borax.

P15

Ln 458       pointed “out” here,

Ln 465    delete “as”

Ln471   “self-healing”       on “the” Diels

P16

Ln 482 delete “at chain”

Ln 483, “to natural polymers”

P17

Ln 518        delete “can conduct” insert “exhibit”

Ln 531  “was only on the”

Ln 532    delete “referred”   insert “inferred”

Ln 536     “pair of electrons on polymeric chains”

P18

Ln 552     “healing character is provided by the hydrophobic”

P19

Ln 580    delete “the”

Ln 582    delete “carbonizable”

Ln 593      “mechanisms are classified” 

Ln 596    ”yet to be fully investigated.”

Ln 597      “for the fabrication of self healing matrixes.”

Ln 603   “Preparaing such electro”                       in “the” form of porous

Author Response

This is a well organized paper reviewing a timely topic, the synthesis of self healing electrically conductive hydrogels.  These are seeing many new emerging applications, especially in tissue engineering.

Comments and Edits.

Page 2  Line 71  It would help the reader to add a brief paragraph explaining the statement  “healing process unlike other self-healing mechanisms”.  Could the author expand on this?

Answer: The statement was explained and supported by references: Typically, selfhealing efficiency of non-conductive hydrogels attains up to 90% [1], 80% [2], or less of the original mechanical strength.

  1. Lei, J.; Li, X.; Wang, S.; Yuan, L.; Ge, L.; Li, D.; Mu, C. Facile Fabrication of Biocompatible Gelatin-Based Self-Healing Hydrogels. ACS Appl. Polym. Mater. 2019, 1, 1350–1358, doi:10.1021/acsapm.9b00143.
  2. Hussain, I.; Sayed, S.M.; Liu, S.; Yao, F.; Oderinde, O.; Fu, G. Hydroxyethyl cellulose-based self-healing hydrogels with enhanced mechanical properties via metal-ligand bond interactions. Eur. Polym. J. 2018, 100, 219–227, doi:10.1016/j.eurpolymj.2018.01.002.

P 8  Ln 237   “Then, a confusion can happen…”   This sentence is not clear, could the authors expand on this, as to what they mean.

Answer: The sentence was rephased to highlight that metal ions are not conductive media. “Then, a confusion can happen when metal ions are considered as conductive media. Typically, authors measure metal salts effect on ionic conductivity by assembling a voltaic cell for measurements and results come by Siemens units.”

P 11 Ln 337  “Contrary to previous work”.  Could the authors, in this paragraph make a few comments as to why electrical conductivity is important for some tissue engineering applications?

Answer: Potentials of the electrically conductive hydrogels is highlighted in the text “Soft electrically conductive hydrogels provide new level of control over biomaterials applied into the human body especially in nervous and cardiac tissue engineering where conducting electricity is a key of successful function [3].”

[3] Distler, T.; Boccaccini, A.R. 3D printing of electrically conductive hydrogels for tissue engineering and biosensors – A review. Acta Biomater. 2020, 101, 1–13, doi:10.1016/j.actbio.2019.08.044.

 General Edits

P1  ln 36   insert “are” before “made”       systems, contain

P1 ln 39 delete molecular, insert dry

P1 ln 44     delete its, insert their

P2

Ln 45 delete has been, insert   emerged “a” few

Ln 47      change Biopolymer to Biopolymers

Ln 53       change hydrogels to hydrogel     open to opens              insert “that” spans

Ln 58        change is to are

Ln 73     insert “that” spans

P3

Ln 100    delete “an”

Ln 112   delete “a”

Ln 117   conduc”t”ive   add t

P4

Ln 135   only “a” few  add a

Ln 136   mostly  to “most”

Ln 143    change “as” to an

Ln 152        insert    purification “of” PANI

P5

Ln 170    insert  requires “a” doping

P6

Ln 177 replace “perform” with “obtain”

Ln 206   delete “a”

P7

Ln 226   delete “been”        and great attention”, in order” to obtain

P8

Ln 251  insert there is “an” additional

P9

Ln 276   replace while with “with”

Ln 284   should read “increasing electrical conductivity was obtained” by

Ln 301   “from” should be “on”

P12

Ln 399  should read “So far there are only a few”

Ln 406  replace  “is” with are

Ln 407   replace “for” with of

Ln 413  Imine bond is “a” very

P14

Ln 433     to point “out” here

Ln 449- ln 450   should read “ prepare a modified polyol for reaction with borax.

P15

Ln 458       pointed “out” here,

Ln 465    delete “as”

Ln471   “self-healing”       on “the” Diels

P16

Ln 482 delete “at chain”

Ln 483, “to natural polymers”

P17

Ln 518        delete “can conduct” insert “exhibit”

Ln 531  “was only on the”

Ln 532    delete “referred”   insert “inferred”

Ln 536     “pair of electrons on polymeric chains”

P18

Ln 552     “healing character is provided by the hydrophobic”

P19

Ln 580    delete “the”

Ln 582    delete “carbonizable”

Ln 593      “mechanisms are classified” 

Ln 596    ”yet to be fully investigated.”

Ln 597      “for the fabrication of self healing matrixes.”

Ln 603   “Preparaing such electro”                       in “the” form of porous

Answer: We appreciate the precision work of the reviewers and we corrected the revised manuscript accordingly

Reviewer 2 Report

The manuscript submitted to IJMS entitled "Irreversible and Self-healing Electrically Conductive Hydrogels Based on Natural Polymers" by Ahmed Ali Nada and co-workers presents a concise short review with an adequate number of references and information on conductive hydrogels based on natural polymers. The clear majority of relevant literature about this subject is referenced and properly discussed by the authors. However, the subject seems off-topic on the aim and scope of the journal where this revision is submitted, and it feels that a different journal could be more appropriate.

Nevertheless, a few minor changes should be made before publication. For instance:

1) A few typos can be found throughout the text.

Line 99: “Insulting” is this what the authors wanted to write?

Line 145: “easy of synthesis” should read “synthesis ease”?

Perhaps the manuscript should be proofread for typos and misspellings.

2) Section 3 should have a table comparing the most relevant information (e.g. polymers used, the conductive medium and conductivity).

3) Figures 12 and 13 have low quality and should be replaced with better quality figures.

Author Response

Comments and Suggestions for Authors

The manuscript submitted to IJMS entitled "Irreversible and Self-healing Electrically Conductive Hydrogels Based on Natural Polymers" by Ahmed Ali Nada and co-workers presents a concise short review with an adequate number of references and information on conductive hydrogels based on natural polymers. The clear majority of relevant literature about this subject is referenced and properly discussed by the authors. However, the subject seems off-topic on the aim and scope of the journal where this revision is submitted, and it feels that a different journal could be more appropriate.

Answer: The Review was submitted to the Special Issue “Polymers from renewable resources” thus fitting well the topic.

Nevertheless, a few minor changes should be made before publication. For instance:

1) A few typos can be found throughout the text.

Line 99: “Insulting” is this what the authors wanted to write?

Answer: Text was revised and the misleading term was changed to “non-conductive”.

Line 145: “easy of synthesis” should read “synthesis ease”?

Answer: corrected

Perhaps the manuscript should be proofread for typos and misspellings.

Answer: the manuscript have been revised thoroughly and carefully for typos and grammar mistakes.  

2) Section 3 should have a table comparing the most relevant information (e.g. polymers used, the conductive medium and conductivity).

Answer: A new Table summarizing described irreversible hydrogels was added to the revised version in Section 3.1

3) Figures 12 and 13 have low quality and should be replaced with better quality figures.

Answer: Figure are replaced with high resolution image (Fig. 12) and redrawn (Fig.13)

Reviewer 3 Report

In this review article, electrically conductive hydrogels based on natural polymers have been discussed with attention to the irreversible and reversible (self-healing) cross-linking properties. The manuscript is well-written by considering recent literature on the topic. Therefore it can be accepted for publication after considering my little concern as in the following-

I wonder how significant is the term ‘natural polymers’ in the title, because the manuscript described polymers obtained from nature (e.g. cellulose) as well as synthetic polymers (e.g. polyaniline) where the monomers were obtained from nature. In the 2nd case, the polymers couldn’t be called as natural polymers, where it would be good to call them as bio-based polymer.  

Author Response

Comments and Suggestions for Authors

In this review article, electrically conductive hydrogels based on natural polymers have been discussed with attention to the irreversible and reversible (self-healing) cross-linking properties. The manuscript is well-written by considering recent literature on the topic. Therefore it can be accepted for publication after considering my little concern as in the following-

I wonder how significant is the term ‘natural polymers’ in the title, because the manuscript described polymers obtained from nature (e.g. cellulose) as well as synthetic polymers (e.g. polyaniline) where the monomers were obtained from nature. In the 2nd case, the polymers couldn’t be called as natural polymers, where it would be good to call them as bio-based polymer.  

Answer: We thank reviewer for this comment. Originally the term “natural polymers” in title was used to inform that the reviewed hydrogels are made of natural polymers as a non-conductive substrate. However, as the reviewer suggested, using the term “bio-based polymers” can cover both the non-conductive substrate and the synthetic polymers from potentially renewable monomers. Therefore, the Title was changed to “Irreversible and Self-healing Electrically Conductive Hydrogels made of Bio-based Polymers”